# The Therapeutic Landscape for *KRAS*-Mutated Colorectal Cancers

**DOI:** 10.3390/cancers15082375

**Published:** 2023-04-19

**Authors:** Simon Manuel Tria, Matthew E. Burge, Vicki L. J. Whitehall

**Affiliations:** 1Conjoint Gastroenterology Laboratory, QIMR Berghofer Medical Research Institute, Herston, QLD 4006, Australia; matthew.burge@health.qld.gov.au; 2School of Medicine, The University of Queensland, Herston, QLD 4029, Australia; 3Department of Medical Oncology, Cancer Care Services, The Royal Brisbane and Women’s Hospital, Herston, QLD 4029, Australia; 4Department of Medical Oncology, The Prince Charles Hospital, Chermside, QLD 4032, Australia; 5Conjoint Internal Medicine Laboratory, Pathology Queensland, Queensland Health, Brisbane, QLD 4006, Australia

**Keywords:** Kirsten rat sarcoma virus (KRAS), mitogen-activated pathway kinase (MAPK), guanine diphosphate (GDP), guanine triphosphate (GTP), guanine nucleotide exchange factor (GEF), GTPase-activating protein (GAP)

## Abstract

**Simple Summary:**

More than a third of all colorectal cancers have a *KRAS* mutation. The complex biology of these cancers has challenged the development of direct targeting inhibitors. Substantial leaps have been made in recent years, with a wave of new generation inhibitors able to specifically target the G12C KRAS mutation. Particularly, Adagrasib and Sotorasib are of note, with several other molecules in development, with the goal of expanding the targets to other KRAS mutations. Therapeutic regimens are being developed to address the emergence of resistance to these inhibitors, including pan-pathway inhibition of adjacent and/or downstream/upstream signaling, blocking metabolic pathways and sensitizing to immunotherapy. The utilization of these other therapies in conjunction with the direct inhibitors provides a significant step forward in the treatment of *KRAS* mutated colorectal cancers.

**Abstract:**

Colorectal cancer is one of the world’s most prevalent and lethal cancers. Mutations of the KRAS gene occur in ~40% of metastatic colorectal cancers. While this cohort has historically been difficult to manage, the last few years have shown exponential growth in the development of selective inhibitors targeting KRAS mutations. Their foremost mechanism of action utilizes the Switch II binding pocket and Cys12 residue of GDP-bound KRAS proteins in G12C mutants, confining them to their inactive state. Sotorasib and Adagrasib, both FDA-approved for the treatment of non-small cell lung cancer (NSCLC), have been pivotal in paving the way for KRAS G12C inhibitors in the clinical setting. Other KRAS inhibitors in development include a multi-targeting KRAS-mutant drug and a G12D mutant drug. Treatment resistance remains an issue with combination treatment regimens including indirect pathway inhibition and immunotherapy providing possible ways to combat this. While KRAS-mutant selective therapy has come a long way, more work is required to make this an effective and viable option for patients with colorectal cancer.

## 1. Introduction

Colorectal cancer is one of the most common forms of cancer and causes of cancer related death globally. Molecularly diverse subtypes influence prognosis and response to therapy [1,2]. The rat sarcoma virus (*RAS*) oncogene is frequently mutated in colorectal cancer with its most frequent isoform, Kirsten RAS (KRAS), influencing the mitogen-activated protein kinase (MAPK) pathway [3].

The MAPK pathway regulates cellular survival, proliferation, apoptosis and differentiation [4]. This pathway allows for the transduction of extracellular signals through the cell to influence gene expression. There are several distinct MAPK signaling pathways. RAS predominantly signals through the extracellular signal regulated kinase (ERK) pathway, but also acts through the phosphatidyl-inotisol-3 kinase (PI3K) pathway. Cancer cells can exploit these systems to increase proliferation, evade apoptosis, and affect surrounding cells to support malignant transformation.

## 2. RAS Signaling

The MAPK and PI3K pathways are interconnected to allow for redundancy to ensure continued cellular function. Figure 1 provides an overview of the complexity of these pathways. It highlights that the RAS protein, while a key signaling component of the MAPK pathway, also has a role in the PI3K/Protein kinase B (AKT) pathway. Signaling is initiated by the binding of a ligand to its receptor tyrosine kinase (RTK) such as the epidermal growth factor receptor (EGFR). This then signals a guanine nucleotide exchange factor (GEF) (e.g., son of sevenless (SOS) protein) to cycle the guanosine diphosphate (GDP) bound to RAS for a guanosine triphosphate (GTP) [5]. This triggers a cascade of phosphorylation which in turn alters the cellular phenotype.

Conformational changes drive the activity of signaling proteins. Discovered in 1990, Milburn et al. identified the Switch I and Switch II regions located on the RAS protein. These regions were proximal to the guanine nucleotide binding region and directly controlled the GDP and GTP exchange of RAS [6]. The cycling of GDP for GTP is an important step that induces a conformational change in RAS allowing it to interact with downstream proteins such as RAF and PI3K for subsequent signaling. This is regulated by the intrinsic GTPase activity within RAS, which is catalyzed by a GTPase-activating protein (GAP), to hydrolyze GTP into GDP and convert RAS back into its inactive form. Oncogenic *KRAS* mutations dysregulate these processes by preventing key structural changes that affect GTP hydrolysis, trapping RAS in a constitutively active state. Mutations including the substitution of glycine at codon 12 into aspartate (G12D), cysteine (G12C) and valine (G12V) directly impede the intrinsic GTPase activity of RAS, while the substitution of threonine at codon 20 to alanine (T20A) prevents GAP recognition [6]. The prevalence of these mutations in the world vary widely but mutations of codon 12 in exon 2 appear to be the most common [7]. Resultant hyperactivity of RAS leads to uncontrolled cell division.

Distinct RAS mutations differentially affect the biochemical and structural properties of the RAS protein, leading to varied signaling perturbation. For example, the G12R, Q61H and Q61L *KRAS* hotspot mutations exhibit up to an 80-fold decrease in intrinsic GTPase activity compared to wildtype KRAS. The G12D, G12V and G13D mutations have less effect, and G12C has minimal effect on GTPase activity [8]. When analyzing GAP-mediated hydrolysis, G12C actually had a lower hydrolysis rate than wildtype KRAS, while G12A and Q61H mutations had up to a 25-fold increase in hydrolysis. In terms of downstream effector recognition of RAF, G12D, G12V and G12R had the least affinity for RAF, while G13, Q61, G12A and G12C mutants had slightly less affinity compared to wildtype RAS. The mechanisms for these differences and how each mutation affects KRAS protein structure and function is beyond the scope of this review but is thoroughly examined by Hunter et al. [8].

## 3. Prognostic Implications of *KRAS* Mutation

Previously, studies have conflicted in regard to the impact of *KRAS* mutation on prognosis in colorectal cancer [9]. This has been settled with multiple recent studies concluding that patients with a *KRAS*-mutant colorectal cancer (25–52%) do in fact have significantly worse prognosis compared to those with wildtype *KRAS* [10,11,12]. However, individual *KRAS* mutations influence outcome differently. A study of 200 patients with Stage I-III colorectal cancer found that the G12V and G12C mutations are associated with worse recurrence free survival [13]. In the metastatic setting, one study (n = 392 patients) found that G12C and G12V mutations were both independent markers of worse overall survival (OS), while another study (n = 1262 patients) observed this relationship only with the G12C mutation [14,15]. A larger retrospective analysis of five metastatic studies (n = 1239 patients) found that G12C mutations are associated with a significantly worse OS and G13D mutations with worse progression free survival (PFS) [10]. These studies highlight that G12C KRAS mutations do in fact confer worse prognosis.

*KRAS* mutation status also affects treatment response. Cetuximab is a current standard of care therapy that targets the EGFR receptor to competitively inhibit the MAPK pathway [16]. This therapy has enhanced patient survival in the metastatic setting, but only in the context of wildtype RAS. That is because *KRAS* mutation confers an intrinsic resistance to this therapy as it leads to constitutive MAPK activation, which is downstream of the EGFR receptor [17]. FOLFOX is another standard of care utilizing a combination of 5-Fluouracil and Oxaliplatin, both creating systemic effects targeting DNA rather than the targeted mechanism of Cetuximab. In a study with 148 patients with metastatic colorectal cancer, it was found that after treating with FOLFOX, patients with *KRAS* mutations had worse PFS than wildtype patients [18]. Particularly, the G12D mutation correlated with worse PFS while the G12S mutation correlated with worse OS [18].

## 4. Direct Inhibition of Mutant KRAS

The targeting of KRAS has been extensively investigated since its discovery as an oncogene, and more earnestly after the discovery that *KRAS*-mutant cancers are resistant to anti-EGFR therapy [19,20]. This makes it an ideal target as it would expand the patient demographic who may respond to anti-EGFR therapy. Drug development companies sought to do this by perturbing the structural change that arises from GTP binding. However, there are several challenges that arise from therapeutically inhibiting KRAS that have suggested it to be “undruggable”. One of these challenges is the significantly high affinity of GTP for KRAS and its high intracellular abundance [5]. Furthermore, an important requirement is that only mutant KRAS proteins are targeted, to limit potential toxicity associated with inhibition of wildtype KRAS that is necessary for normal cellular functioning.

There have been several molecules that are currently in development and a few that have progressed to clinical trials. To date, there have been two KRAS inhibitors targeting G12C mutants that are FDA-approved: AMG510 (Sotorasib) and MRTX840 (Adagrasib). There are also several drugs that have proceeded to clinical trial including LY3499446, JAB-21822, JNJ-74699157, GFH925, YL-15293, MRTX-1133 and JDQ443 [21,22,23,24,25,26]. Revolution Medicines has presented a range of drugs that directly target G12C KRAS mutants and others including G12D, G12V and G12X. The direct inhibitors of KRAS described in this review are outlined in Table 1 for ease of reference.

### 4.1. KRAS G12C Inhibition

The first generation KRAS inhibitors selectively targeted G12C mutants by binding the Switch II pocket (SII-P) (Figure 2) [27]. The SII-P is a region that lies between the nucleotide binding pocket that binds GDP and GTP, and the Switch II region. Tethering compounds, electrophiles, and acrylamides were initially screened for preferential binding of KRAS G12C mutants in their GDP-bound state. The aim was to decrease their affinity for GTP and downstream effector proteins, trapping them in their inactive state. ARS-853 was soon developed and excelled in its ability to induce apoptosis, decrease proliferation and inhibit both MAPK and PI3K pathway signaling [28]. The drug was able to engage greater than 90% of available KRAS proteins, a proportion much larger than expected as it can only bind GDP-bound KRAS. This was contradictory to the expectation that *KRAS* mutations prevent GTPase activity and therefore there should be more GTP-bound KRAS and less engagement for ARS-853. The studies explained this by showing that KRAS G12C mutants still exhibit a basal level of GTPase activity without GAP cataclysm to cleave GTP into GDP, allowing mutant KRAS proteins to revert to their inactive form [29]. This study suggested that while KRAS G12C mutants are resistant to upstream EGFR inhibition and to GTP hydrolysis, these activities are still relevant and can be targeted by ARS-853.

While promising, the translation of ARS-853 into the clinic was unsuccessful due to unsatisfactory plasma levels and a short half-life [30]. Structural refinement of ARS-853 led to ARS-1620; in vitro it had superior targeting of KRAS G12C and in vivo it displayed superior half-life and oral bioavailability [30]. Interestingly, the efficacy of this drug in KRAS-dependent cell lines was affected by the culture system, with some lines displaying >20-fold difference in sensitivity under a 3D culture system compared to 2D systems. Tested in vivo, ARS-1620 induced significant tumor growth inhibition and regression selectively with *KRAS* G12C-mutated xenografts with no response in G12V-mutated xenografts. Furthermore, using patient-derived xenografts (PDXs), ARS-1620 achieved similarly significant tumor growth inhibition and regression with those bearing G12C mutations compared to those without.

#### 4.1.1. Sotorasib: First Clinically Available KRAS Inhibitor

AMG510 (Sotorasib) is the first clinically available KRAS-mutant selective inhibitor, able to specifically bind G12C KRAS mutants. Sotorasib was developed by Amgen and approved by the FDA in 2021 for the treatment of KRAS-G12C-mutant non-small cell lung cancer [31,32]. Sotorasib is an improvement upon the ARS-1620 inhibitor; structural changes led to optimized covalent binding of the new inhibitor to the Cys12 residue of KRAS G12C mutants. Particularly, the addition of a benzene ring allowed additional binding to His95, Tyr96 and Glu99 residues, which formed a pocket proximal to the SII-P region of the protein (Figure 2) [33,34]. Compared to ARS-1620 in vivo, Sotorasib showed superior dose and time-dependent induction of tumor regression and MAPK inhibition, to ARS-1620 [33]. Sotorasib also selectively inhibited G12C-mutant KRAS over wildtype or G12V variants in vivo, including in human cell line xenografts, PDX models and a syngeneic mouse model. Sotorasib was the first selective inhibitor of G12C-mutant KRAS to reach Phase I and Phase II clinicals trials, demonstrating anti-tumor activity, with tolerable toxicity and safety parameters [35,36]. All clinical trials for Sotorasib and subsequent drugs are outlined in Table 2.

#### 4.1.2. Adagrasib

MRTX849 (Adagrasib) is the second KRAS-G12C-mutant selective inhibitor to be approved by the FDA, indicated for the treatment of KRAS G12C-mutated NSCLC [37]. Adagrasib inhibits MAPK signaling by covalently binding to the SII-P region of GTP-bound KRAS G12C mutants (Figure 2). In doing so, Adagrasib reduces cell viability at nanomolar concentrations. In vivo, Adagrasib displayed a particularly strong dose-dependent inhibition of pERK and subsequent tumor regression [37]. Using the MIA PaCa-2 pancreatic cancer cell line in a xenograft model, Adagrasib was able to induce complete tumor regression after 10 days, while Sotorasib required 20 days to reach only near complete tumor regression at the same dose [33,37]. Furthermore, this occurred with Adagrasib treatment being initiated when tumor size was roughly double that of when Sotorasib treatment was started. Although results may be influenced by specific study differences, Adagrasib appears to be more effective at inducing tumor regression, but more work is required to definitively conclude this and if so, what the mechanism is.

Approval for Adagrasib came after the Krystal-1 Phase I clinical trial that showed an objective response rate of 43% in patients with locally advanced or metastatic NSCLC [38]. Adagrasib was administered to 112 patients, in combination with platinum-based chemotherapy or immune checkpoint therapy. With only eight patients observed significant adverse reaction warranting discontinuation, there was an objective response rate of 42.9% and a disease control rate of 79.5%.

#### 4.1.3. ASP2453

ASP2453 is a recently developed KRAS inhibitor that similarly binds the Cys12 residue of mutant KRAS [39]. ASP2453 selectively inhibits cell growth and downstream binding of GEF and RAF proteins for MAPK, cell cycle and PI3K/AKT activation at nanomolar concentrations. Tested in vivo, ASP2453 induces significant tumor regression at the highest tested concentration 30mg/kg, which was well tolerated. ASP2453 synergized with Erlotinib (EGFRi), Trametinib (MEKi) and anti-PD1 antibodies, all inducing significant tumor regression greater than single agents.

Compared to the FDA-approved Sotorasib, ASP2453 was found to be more potent and sustained a longer response [39]. In vivo, ASP2453 displayed higher potency by inducing near complete tumor regression of MIA PaCa-2 xenografts at roughly a third the dose for Sotorasib. Furthermore, after initial treatment with Sotorasib, mice switched to ASP2453 showed significantly greater tumor regression compared to mice who had increased doses of Sotorasib. In an experiment comparing ASP2453 with Adagrasib in vivo, ASP2453 treatment reached near complete tumor regression at a dose >3 fold lower than the dose for Adagrasib in a subcutaneous xenograft model of MIA PaCa-2 [37,39]. This suggests ASP2453 has improved potency over both FDA-approved drugs Adagrasib and Sotorasib.

The KRAS inhibitors ASP2453, Sotorasib and Adagrasib are the most recently developed KRAS inhibitors with comprehensive published results in vitro and in vivo on their ability to selective inhibit KRAS G12C mutated cancers and downstream pathways. Adagrasib and Sotorasib are the only two of those that have entered clinical trials and have received FDA approval. Several other KRAS G12C inhibitors are currently being investigated in clinical trials: JAB-21822, GFH925 and YL-15293. The trial of LY3499446 was terminated due to unexpected toxicity, and JNJ-74699157 was terminated due to skeletal muscle toxicities and lack of efficacy [21,22,24,25,26].

#### 4.1.4. JDQ443

Novartis has developed JDQ443, a KRAS G12C specific inhibitor with unique binding to KRAS in comparison to both Adagrasib and Sotorasib [40]. JDQ443 has progressed into clinical trials after promising preclinical results. The preclinical studies showed KRAS G12C-specific inhibition in vitro and in vivo with dose-dependent MAPK signaling downregulation. However, discussed in a later section, there are already known mutations that confer resistance to both Sotorasib and Adagrasib, particularly mutations of the SWIIP that prevent normal binding. The activity of JDQ443 in the presence of these mutations is varied; mutations of the Histidine 95 residue such as H95R, H95Q and H95D are shown to reduce sensitivity to JDQ443 but still reduce proliferation and signaling. However, other mutations such as R68S, Y96C and Y96D confer complete resistance in reducing proliferation and signaling. While JDQ443 also binds to the Cys12 residue in the SWIIP, its structure allows different binding interactions to the pocket in comparison with Sotorasib and Adagrasib and explains why it can be effective in some but not all mutations. Interestingly, researchers found that JDQ443 alone was not more effective in reducing tumor volume in vivo compared to Sotorasib and Adagrasib in different G12C-mutated cancer models. However, when in combination, JDQ443 had better synergy with Ribociclib (CDK4i), TNO155 (SHP2 inhibitor) and Trametinib compared to Sotorasib and Adagrasib both in vitro and in vivo.

JDQ443 is currently being evaluated in several clinical trials termed the KontRASt series trials, either alone or in combination with other drugs in KRAS G12C solid cancers. [41,42,43,44]. The KontRASt 01 and 03 trials are Phase 1b/2 trials looking at JDQ443 alone or in combination with TNO155, Tislelizumab, Ribociclib or Cetuximab in different KRAS G12C-mutated advanced solid tumors. KontRasT-02 is a Phase III trial evaluating JDQ443 with or without Docetaxel in NSCLC exclusively. To date, two patients from KontRASt-01 have been published to show a partial response with tumor regression of NSCLC primary tumors and duodenal papillary cancer secondary liver metastases.

### 4.2. Alternate KRAS-Mutant Inhibition

Revolution Medicines has recently demonstrated a new strategy for targeting KRAS mutations, presenting inhibitors that can bind GTP-bound KRAS, termed KRAS(ON) [45,46,47]. These inhibitors bind both a Cyclophilin A protein and the KRAS protein, creating a tri-complex structure to collectively inhibit signaling. Contrasting to previously developed inhibitors, the tri-complex inhibitors are able to target KRAS proteins that are constitutively activated, rather than limiting the mechanism of activation. Furthermore, Revolution Medicines has been the first to target non-G12C KRAS mutants, developing G12D and G12V selective inhibitors, as well as a non-selective, multi-RAS targeting drug.

RMC-6291 targets KRAS G12C mutated proteins. Validated in vitro, RMC-6291 can inhibit MAPK signaling at a faster rate and with higher potency compared to Adagrasib and Sotorasib. In vivo, RMC-6291(200 mg/kg) resulted in a 68% Objective Response Rate (ORR), while the Adagrasib (100 mg/kg) treatment resulted in 43%, in a range of NSCLC KRAS G12C-mutated cancers. RMC-6291 has an improved rate and durability of response compared to Adagrasib. Furthermore, by targeting the active state, RMC-6291 inhibits cell lines in vitro that have secondary mutations that confer resistance to both Adagrasib and Sotorasib. RMC-036 and RMC-037 have been developed to target the KRAS G12D mutations with promising in vivo results, inducing complete tumor regression of a KRAS G12D-mutated pancreatic xenograft [45].

RMC-6236 is the first multi-targeting KRAS inhibitor against G12C, G12D, G12V and G12X-mutated KRAS cancers. Broad-spectrum activity is seen in pancreatic, colorectal and NSCLC type cancers, presenting an exciting new therapeutic option for *KRAS*-mutated cancers. Interestingly, similar to Sotorasib and ASP2453, RMC-6236 synergized with anti-PD1 therapy in a colorectal tumor model. RMC-6236 treatment was shown to increase the population of CD8+ T-cells and decrease M2 macrophages and myeloid driven suppressor cells in the tumor environment. The combination of anti-PD1 and RMC-6236 induced a complete response rate in all mice, compared to 2/10 mice for anti-PD1 alone and 6/10 for RMC-6236 alone.

A recent publication discovered a potentially new avenue for KRAS inhibition through non-covalent, reversible binding of the SII-P region [48]. This non-covalent, reversible binding allowed inhibitors to bind mutants outside their designed specificity. These findings, initially evaluated in a cell-free system, showed Sotorasib, Adagrasib and MRTX-EX185 (G12D specific drug) were able to bind Wt KRAS, KRAS (G12D) and Wt HRAS. Moving in vitro, using their own developed BRET assay, they found that Adagrasib and MRTX-EX185 were able to engage several other KRAS mutant proteins, including those with mutated glycine at codon 12 and 13, mutated glutamine mutated at codon 61 and even BRAF V600E mutants. Meanwhile, Sotorasib remained to have specificity for only G12C mutatant proteins. However, the IC50 values for these lines were several magnitudes higher (micromolar range) than that of the KRAS G12C-mutated lines (nanomolar range). This study revealed that while these inhibitors are developed for very specific KRAS mutants, their activity may not be limited to those mutants.

MRTX1133 is another inhibitor that has been developed and published quite recently by Mirati Therapeutics, the developer of Adagrasib [49]. This KRAS inhibitor is unique in its ability to directly target the Asp12-mutated residue of G12D-mutated KRAS proteins. This inhibitor, modeled after the initial G12C inhibitor Adagrasib, inhibited several G12D-mutated cell lines in vitro at nanomolar concentrations. In vivo, MRTX1133 displayed robust, dose-dependent tumor regression and repression of Erk signaling. While the researchers did contrast their inhibitor against a single KRAS WT cell line, showing it was significantly less effective, this was only conducted in vitro and more work is required to determine the selectivity of their drug against other KRAS mutants and wildtype background. However, their work highlights an important need in making KRAS-selective inhibitors accessible to a wider population. MRTX-1133 development has also progressed into clinical trials, and is the first inhibitor targeting G12D KRAS mutants to do so (Table 2).

## 5. KRAS G12C Inhibitor Resistance

Despite the novelty and significance of being able to directly target *KRAS*-G12C-mutant cancers, clinical trials of both Adagrasib and Sotorasib have shown acquired resistance. The mechanisms of resistance include acquisition of mutations in the binding pockets targeted by the drugs, secondary activating mutations in *KRAS*, pathogenic mutations in other members of the MAPK pathway, and oncogenic gene rearrangements/fusions and amplification of the *KRAS* G12C allele or the MET gene [50,51]. In particular, the activating mutation of the *RAF* can occur directly downstream of RAS [51,52].The upregulation of KRAS suppressor protein 1 (KSR1) has been identified as a mechanism for resistance to Sotorasib [53]. KSR1 was found to act as a scaffolding protein that can bypass KRAS and continue signaling despite inhibition of KRAS G12C [53,54].

Acquisition of mutations affecting the KRAS cysteine residue, or the SII-P binding region, significantly affected the binding of both Adagrasib and Sotorasib to confer resistance. However, mutations additional to the original *KRAS* G12C mutations, such as G12D, G12V, G13D, and G12R, were also found that further activated KRAS after Adagrasib treatment. Furthermore, acquired mutations in the MAPK pathway were also significantly associated with resistance, particularly in *NRAS* (an isoform of RAS), *MAP2k1*/*MEK1* and of *EGFR*.

Combination therapy targeting multiple pathways may combat these resistance mechanisms. Adagrasib has been shown to synergize with several drugs that target upstream and downstream molecules, including HER2 (Afatinib), SHP2 (RMC-4450), mTor (Vistusertib), CDK4/6 (Palbociclib) [37]. Validated through a panel of KRAS-dependent and -independent cell lines in vitro and in vivo, the combination of these respective drugs with Adagrasib exhibited significant dose-dependent tumor regression compared to single agents. The efficacy and safety of Adagrasib has recently been evaluated in a Phase I/IB clinical trial of patients with solid tumors harboring a *KRAS* G12C mutation [55]. Adagrasib was found to be sufficiently tolerated and induced a roughly 50% partial response in NSCLC (8/15) and colorectal cancer (1/2) patients. Several other Phase 1, 2, and 3 trials are further evaluating the efficacy and safety of single agent Adagrasib and in combination with Docetaxel, Palbociclib, Cetuximab, BI 1701963, and TNO155 in *KRAS* G12C mutated cancers [56,57,58,59].

Sotorasib has similarly been demonstrated to be effective in combination therapies. There are several NSCLC Phase 1 and 2 trials evaluating combinations with bevacizumab (VEGF inhibitor), VS-6766 (MEK inhibitor), RMC-4630 (SHP2 inhibitor), Cisplatin/Carboplatin and Pemetrexed, as well as a Phase 3 trial of combination Sotorasib with Docetaxel [60,61,62,63,64,65,66,67]. For colorectal cancer, CodeBreak 300 is an actively recruiting Phase 3 clinical trial for refractory metastatic cancer, evaluating Sotorasib in combination with Panitumumab (EGFR inhibitor) versus either TAS-102 or Regorafenib [68].

## 6. Indirectly Targeting KRAS-Mutant Cancers through Pan-Pathway Inhibition

The direct targeting of KRAS-mutant proteins is a powerful tool but is not the only option for treating *KRAS*-mutant cancers. The inhibition of the other MAPK pathway components can indirectly target KRAS-mutated cancers. Figure 3 illustrates other potential targets, particularly involving the MAPK, PI3K, and JAK/STAT pathways. One such member is the MEK protein, which is downstream of RAS in the MAPK pathway. The drugs that target MEK have been extensively investigated with some already approved by the FDA for melanoma, neurofibroma, thyroid carcinoma, and NSCLC [69]. However, these drugs have demonstrated limited benefit as monotherapy for *KRAS*-mutated colorectal cancers, due to acquisition of resistance [70,71,72]. Furthermore, it was discovered that MEK inhibitors can have an unintended secondary effect of activating the WNT pathway, another driver of colorectal tumorigenesis [73]. While MEK inhibitors can reduce proliferation and growth of cells, they can also induce a stem-like phenotype that can increase cell survival. This secondary effect is hypothesized to be one of the causes of MEK resistance. Another mechanism for resistance to MEK inhibition is the activation of the PI3K pathway [74,75]. By combining inhibition of MEK with PI3K inhibition, the resistance is overcome and results in cell death in vitro and tumor growth regression in vivo [75]. Furthermore, introducing EGFR inhibition to the MEK and PI3K combination increased in vivo efficacy in a patient-derived xenograft model [76]. However, in the clinic the combination of MEK and PI3K inhibitors showed little efficacy with varying tolerability, with most reports showing the combination is not well tolerated [77,78,79,80,81,82].

The JAK1/2-STAT pathway was found to be upregulated after MEK inhibition in *KRAS*-mutated colorectal cancers [70]. This compensatory mechanism was found to work through an axis that involved Erk, c-Met, ADAM17 and Jak/Stat. Synergistic growth inhibition was observed in vivo following combination of MEK inhibition with either JAK or c-Met inhibition. Similarly, combined ADAM17 and MEK inhibition led to greater tumor growth inhibition in *KRAS*-mutated lung cancer, compared to MEK inhibition alone [83]. Mechanistically, it was found that this worked through the PI3K/mTOR pathway rather than the JAK/STAT pathway, in both the previously mentioned lung cancer model and a breast cancer model (although not *KRAS*-mutant) [83,84]. The combination of c-MET and MEK inhibition has been trialed clinically without promising results [85]. The trial reported that while certain arms were completed, several toxicities, tolerability issues, and a lack of clinical response showed that preclinical data did not translate into clinic [86].

It is known that EGFR inhibition is not effective in *KRAS*-mutant colorectal cancers, due to downstream activation of the pathway. Therefore, targeting a downstream component of this pathway will likely increase the response to EGFRi. This was studied in a Phase Ib/II clinical trial, evaluating MEK162 (MEKi) with Panitumumab (EGFRi) combination therapy in mCRC stratified by *RAS* mutation status and prior treatment with EGFRi [87]. The study reports that while there were no reported dose limiting toxicities, all the participants displayed treatment adverse effects, with serious adverse effects in 30–62.5% of participants. In the phase 2 treatments, only the group with wildtype *RAS* tumors with prior EGFRi therapy achieved an objective response rate (6.7%). Furthermore, both groups with or without prior EGFRi therapy with wildtype *RAS* have higher PFS and OS compared to the groups with mutated *RAS*.

The MAPK pathway can activate the cell cycle through Cyclin D1, which complexes with cyclin-dependent kinase 4 and 6 (CDK4/6), resulting in activation of downstream effectors of the cell cycle. Combination MEK and CDK4/6 inhibition has been shown to be synergistically reduce *KRAS*-mutant colorectal cancer growth both in vitro and in vivo [88]. Five clinical trials have already begun evaluating this combination in *KRAS*-mutated solid cancers, with data to date suggesting this is a promising avenue for indirect targeting of KRAS-mutated cancers [89,90,91,92].

BI-3406 is an inhibitor of SOS1, which is effective at nanomolar concentrations and sufficiently antagonizes GEF to dampen MAPK pathway activity in *KRAS*-mutated cancers [93]. Tested across a range of colorectal, lung, skin, and pancreatic cancer cell lines in vitro, BI-3406 displays *KRAS*-mutant cancer specific activity, capable of inducing apoptosis through MAPK inhibition. In vivo, Bi-3406 induces significant tumor growth regression and MAPK inhibition in a range of colorectal, lung, and pancreatic xenograft models without toxicity [93]. Bi-3406 also synergizes with MEKi to induce significant tumor regression in *KRAS*-mutant cell line xenografts of pancreatic and colorectal cancer and of PDX models of colorectal cancer. This SOS1 inhibitor can also enhance MAPK suppression in combination with the KRAS inhibitor Sotorasib in vitro [93].

## 7. Exploiting the Altered Metabolic Pathways to Target KRAS-Mutant Cancer

### 7.1. Ferroptosis

β-elemene is a naturally occurring herbal compound found to induce ferroptosis, an iron-induced form of cell death [94]. The combination of β-elemene and Cetuximab significantly inhibited proliferation and induced cell cycle arrest and cell death in vitro, in comparison to either agent alone. Tested in vivo, β-elemene alone significantly induced tumor regression, which was further improved by combination with Cetuximab. Further work, however, is required to fully understand the mechanism behind this process. RAS-selective lethal 3 (RSL3) was also found to induce ferroptosis in *KRAS*-mutant cells, which is also enhanced by Cetuximab to increase killing [95]. Mechanistically, Cetuximab was found to activate p38 of the MAPK pathway, which subsequently inhibited the Nrf2/HO-1 signaling molecule moiety, which is important for the regulation of oxidative stress in cells. In this way, Cetuximab directly increases the formation of lethal reactive oxidation species (ROS) to induce death. These two studies highlight possible alternative routes for *KRAS*-mutant cancer treatment beyond small molecule inhibitors.

### 7.2. Glucose Metabolism

Cancer cells are known to undergo the Warburg effect, which is an abnormally increased rate of glycolysis to generate ATP for energy [96,97,98]. Vitamin C was found to act in a novel, multi-faceted mechanism to inhibit KRAS-mutated colorectal cancer cells. At high concentrations (5–10 mM), Vitamin C treatment directly quenches intracellular ROS through an unspecified mechanism. In doing so, the co-localization of RAS to the plasma membrane is affected and downstream expression and phosphorylation of ERK1/2 and MEK1 is prevented. Incidentally, this led to a sensitization of resistant cell lines to Cetuximab. Furthermore, by inhibiting the transcription of expression of key proteins in glucose metabolism, PTB1, PKM2, and GLUT1, glucose uptake is dampened and leads to cell death [96]. A separate group had found that Vitamin C in lower concentrations (100uM-3mM) is oxidized into dehydroascorbate, which depletes important proteins that maintain intracellular ROS [99]. The increased ROS directly and indirectly affects the activity of GAPDH, a key player of the glycolysis pathway. Directly, increased ROS oxidizes GAPDH into an efficient state and, indirectly, ROS induces DNA damage that activates PARP and reduces NAD+, the substrate of GAPDH in glycolysis. The effect of these leads to an attack by free ROS and also an inability to produce glucose that leads to death and is amplified by low glucose. The use of Vitamin C has had varied responses in clinical trials; there are currently two trials looking at Vitamin C in colorectal cancer patients with KRAS mutations. In a Phase III study, 442 metastatic patients were treated with either FOLFOX with or without Bevacizumab (Control) or with high dose Vitamin C in combination with FOLFOX with or without Bevacizumab (Experiment) [100]. In this setting, both groups had ~50% rate of KRAS mutation, and while overall the addition of Vitamin C did not significantly increase PFS or PS, and the RAS patients did show significantly increased PFS. A Phase II study is currently recruiting early stage and locally advanced colorectal, lung, and pancreatic cancer patients for treatment for single agent, high dose-Vitamin C.

### 7.3. Glutamine Metabolism

Alongside increased energy demands is the need for amino acids as building blocks for the increased biosynthesis (particularly protein synthesis) needed for cancer [97,98]. A way KRAS mutations are able to support this is through the upregulation of amino acid transporters, including SLC7A5, SLC38A2, and SLC1A5 through the YAP1 transcriptional coactivator, which import amino acids such as glutamine into the cell [101]. SLC7A1 plays a more complex role; acting as an antiporter, it can exchange intracellular glutamine for other amino acids to feed biosynthesis of proteins and fatty acids [102]. This transporter, upregulated by KRAS, was also found to have an effect on the transcription of several pathways including DNA replication, autophagy, and mTorc signaling. Upon deletion, a marked growth reduction in organoids in vitro and tumor growth in vivo revealed the ability to sensitize KRAS-mutated cancers to mTorc inhibitors, Rapamycin, and Everolimus.

SLC25A22 transfers glutamine into the mitochondria for metabolism in the Tri-Carboxylic Acid pathway for the production of energy and renewal of substrates of other pathways. KRAS mutations are found to increase this protein, which leads to downstream repression of DNA transcription through DNA demethylation and CpG hypermethylation [103]. One of the phenotypes of this transcriptional reprogramming is stemness through the Wnt/B-Catenin pathways, which leads to chemotherapy resistance; the blockage of SLC25A22 led to the sensitization to 5-FU in vitro and in vivo. In order to target this pathway, a drug to inhibit glutamine synthesis was developed, Telaglenastat, a Glutaminase inhibitor. With early testing in renal cell carcinoma, a Phase I trial showed safety and suggested efficacy with Telaglenastat. However, in a Phase II trial, it failed to showed efficacy in combination with Cabozatinib, a tyrosine kinase inhibitor [104,105]. In a KRAS-mutant colorectal cancer context, there is a completed Phase1b/2 trial awaiting reports that examined the safety and efficacy of Telaglenastat in combination with Palbociclib [106].

## 8. Immunotherapy Options for *KRAS*-Mutant Cancers

Benefit from immunotherapy is limited to a small subset of colorectal cancers that have deficient DNA mismatch repair (dMMR) activity, resulting in microsatellite instability (MSI) [107]. The benefit is seen in this subtype because of the increased tumor mutational burden in these dMMR cancers that create more neoantigens that can better activate the immune system and induce lymphocyte tumor infiltration [107]. Phase II trials have shown that immunotherapies significantly increase survival and objective response rate in patients with metastatic MSI colorectal cancer [108,109,110]. However, KRAS mutation is less common in the setting of MSI [111], and as was demonstrated in the Keynote-164 trial, *KRAS*-mutated MSI cancers confer a lower objective response compared to *KRAS*-wildtype cancers [110]. Furthermore, the Phase III Keynote-177 trial reported that while Pembrolizumab did not significantly increase OS and PFS in late stage MSI/dMMR colorectal cancers, it was better tolerated [112,113]. This has led to it being approved for use in the first line for advanced/metastatic MSI colorectal cancers [114]. Again, *KRAS*-mutational status was not a significant factor in predicting patient outcome or response to Pembrolizumab [113].

Research efforts are focused on reversing the immunosuppressive microenvironment of non-MSI cancers to enhance response to immunotherapy (summarized in Table 3). For example, the cytokine Interleukin-17A was found to increase PD-L1 immune checkpoint expression and decrease cytotoxic T-cells. Blocking IL-17A sensitized cells to immune checkpoint blockade [115]. Tribbles homolog 3 (TRIB3) was found to decrease T-cell tumor infiltration by activating the EGFR pathway, whilst TRIB3 inhibition sensitized cells to immunotherapy [115,116]. Neutrophil extracellular traps (NETs) is an innate immune response that leads to decreased immunotherapy efficacy, which can be reversed by treatment with DNAse1 [117]. The mitochondrial antiviral signalling protein (MAVS) was found to promote adaptive immune tumor responses through an interferon axis, upregulating both CD8 positive T-cell cytotoxicity and PD-L1 expression [118]. Through the introduction of MAVS expression in cell lines, xenograft models displayed synergistic tumor regression when treated with PD-L1 inhibitors compared to without MAVS. Several chemotherapeutic combinations have also been found to increase immune responses by promoting tumor recognition and infiltration of tumors by adaptive and innate immune cells. These combinations include Mitomycin C plus HSV-1 (attenuated oncolytic virus), a standard regimen in the clinic consisting of 5-FU plus oxaliplatin, and a new combination consisting of Tas-102 plus oxaliplatin [119,120,121]. While these studies were not conducted specifically for *KRAS*-mutated cancers, they do provide significant groundwork in applying immunotherapy to previously resistant cancer subsets.

Bispecific antibodies are a recent development in the immunotherapy arsenal, capable of engaging T-cells with tumor-associated antigens [122]. Expression of guanyl cyclase c (GUCY2C) is limited to gastrointestinal epithelia. PF-07062119 is a bispecific antibody developed to target cancers expressing GUCY2C with dose-dependent efficacy. When contrasted against a non-GUCY2C expressing line, PF-07062119 did not have an effect, and lines that had moderate expression of GUCY2C required a higher dose to reach tumor regression. This effect was seen regardless of *KRAS*-mutational status, showing good efficacy with *KRAS*-mutated cancers, provided they expressed GUCY2C. More importantly, the treatment of PF-07062119 alone had a secondary effect of increasing PDL1 and PD1 expression. Combining suboptimal concentrations of PF-07062119 with anti-PD1 or anti-PDL1 antibodies, there was an enhanced effect of tumor growth inhibition compared to single agents. Furthermore, PF-07062119 also had a synergistic effect when used at suboptimal doses with anti-VEGF therapy, inducing complete tumor regression.

Adoptive T-cell transfer is a form of personalized medicine that utilizes a patient’s own T-cells, expanded ex vivo. There is a current Phase II clinical trial evaluating this form of therapy in a range of solid metastatic cancers including colorectal cancer [123]. There are two patients reported to have had KRAS-mutant-specific T-cells isolated and subsequently treated with [124]. For the first patient, three out of ten metastatic lung lesions were resected from the patient; T-cells specific to the *KRAS* G12D mutation were isolated from the patient and expanded before transferring them back to the patient. Treatment resulted in complete regression of the seven other lesions in just 40 days. However, one lesion did regress, due to a mutation in the HLA molecule the T-cell binds to. In the second patient, T-cell adoptive transfer was unresponsive, possibly due to low KRAS G12D specific T-cell numbers.

Interestingly, Sotorasib has displayed a clear immunogenic benefit, shown to synergize with anti-PD1 therapy [33]. In vivo, treatment with AMG-510 synergized with anti-PD1 therapy to induce significant tumor regression. Furthermore, it also prevented recurrence in mice that had cleared tumors and were re-challenged with tumor cells. Validated using RNA-seq, immunohistochemistry, and in vitro co-culturing systems, Sotorasib was found to increase T-cell priming and subsequent tumor infiltration by promoting an inflammatory microenvironment, antigen presentation, and stimulating the innate immune system [33]. The researchers only presented phenotypic data, showing Sotorasib treatment induced these phenotypes by increasing chemokine secretion, working as a chemoattractant for immune cells, including CD8+ T-cells, dendritic cells, and macrophages. Interestingly, they contrasted these results with mice/cells treated with a MEK inhibitor and found certain differences. The MEK inhibitor could impair T-cell proliferation but not increase immune cell infiltration in tumors, while Sotorasib had no effect on T-cell proliferation but could increase immune cell infiltration. The signaling between RAF and MEK is shared, and it is within reason that the effects of inhibiting either one should be the same, but they are not. This study opens up an interesting line of study as to how the Sotorasib can induce these effects, different to what the MEK inhibitor can do, despite working in the same pathway.

Vaccines provide a way to expose the body’s own immune system to mutant proteins in an attempt to train it to pre-emptively attack cells with these proteins. While this has been explored more comprehensively in KRAS-mutant pancreatic cancer, there has been some success with colorectal cancer [125,126]. Furthermore, this type of intervention has been found to work better when administered post-resection or in combination with chemotherapy and immunotherapy [125]. Current active trials in this field are looking at combining RAS-mutant vaccines with therapies that will enhance this directly, such as with CTLA-4 and PD-1 inhibition [127,128]. These trials are looking at the metastatic setting with MSS cancers that have prior treatment with standards of care.

Significant progress has been made in sensitizing previously resistant models to immunotherapy. Along with immune checkpoint blockade, bispecific antibodies and adoptive T-cell transfer, the promise of immunotherapy may soon be realized for a wider range of patients with colorectal cancer.

## 9. Conclusions

The development of KRAS G12C inhibitors has revolutionized drug development for *KRAS*-mutated cancers. Sotorasib was the first of its kind to enter the clinic, selectively targeting KRAS G12C mutations, with additional immunogenic capabilities. Adagrasib has similar properties but exhibited slightly higher potency and has also gained FDA approval. Both Adagrasib and Sotorasib have performed favorably in the clinic. ASP2453 is the most recent KRAS G12C inhibitor and has performed well in preclinical studies with improved efficacy compared to Sotorasib. There are also several other KRAS G12C inhibitors that have entered clinical trials with varying degrees of success.

A substantial proportion of colorectal cancers harbor non-G12C *KRAS* mutations. Mirati Therapeutics and Revolution Medicines have both developed drugs that can target the G12D mutation in KRAS, with the latter expanding to other mutations. Other methods of targeting *KRAS*-mutated cancers are using pan-pathway inhibitors, most commonly targeting the downstream MAPK molecule MEK, with enhanced efficacy with combination therapy targeting the EGFR, CDK, JAK/STAT, PI3K pathways with a KRAS inhibitor. As these inhibitors largely target the active pathways, their activities are not limited to individual point mutations, potentially treating a larger cohort of patients with a *KRAS*-mutated cancer.

Furthermore, there also lies the possibility of exploiting the altered metabolism of KRAS-mutant cancers against themselves. These cancers upregulate proteins in order to increase energy, protein, and fatty acid synthesis to support their prolonged, uncontrolled growth. By preventing the uptake of starting substrates such as glucose and glutamine, their downstream metabolism is prevented, and the cells are unable to efficiently proliferate. However, these types of therapies, including pan-pathway inhibition, come with a caveat: while the intention is to target cells that have overexpressed these metabolic components, there is always a chance of inducing off-target effect. Balancing these in a way that sufficient therapeutic effect is delivered without subsequent systemic damage is important.

The use of immunotherapy has revolutionized cancer care in recent years. Despite early unfavorable responses in clinics, there is promise that resistant mechanisms in colorectal cancer might be overcome. Several molecules, proteins, and unique drug combinations have been found to increase tumor immune recognition, infiltration, and checkpoint expression. Lastly, KRAS vaccines have gotten some recent traction after relatively slow progress and present another potential therapeutic intervention. While direct KRAS inhibitors have already shown synergistic effects with immunotherapy, these novel methods pave the way to potentially treat a larger cohort of *KRAS*-mutated cancers.

Cancer is an unpredictable disease, with therapeutic resistance being a major challenge. Acquired resistance has already been reported for Sotorasib and Adagrasib. The combination of KRAS inhibitors with chemotherapies and immunotherapies holds enormous promise for improving outcomes for patients with *KRAS*-mutated cancer. Such options are already being evaluated in the clinic and the speed in which these drugs have progressed clinically is truly remarkable.

## Figures and Tables

**Figure 1 cancers-15-02375-f001:**
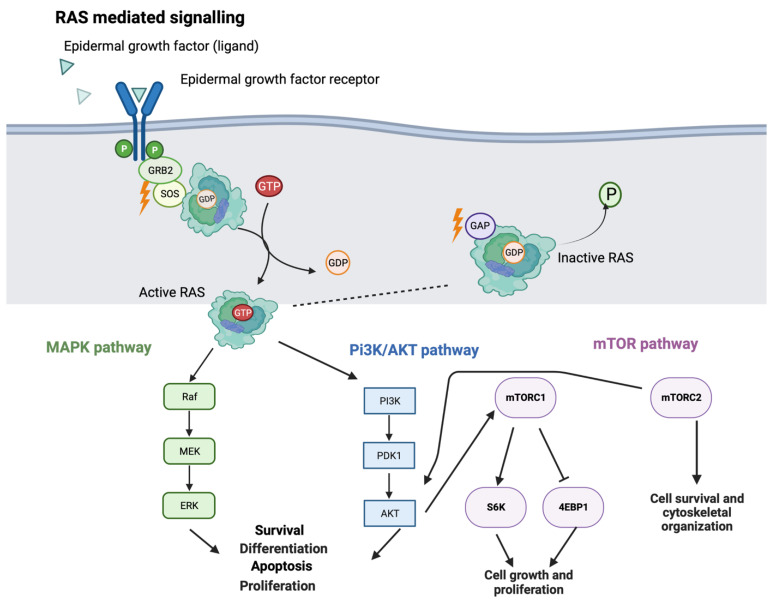
RAS-mediated signaling pathway. Signaling begins with a ligand such as the epidermal growth factor (EGF) binding to a receptor tyrosine kinase such as the EGF receptor (EGFR). This binding activates downstream guanine nucleotide exchange factor (GEF) proteins such as the son of sevenless (SOS) protein to mediate cycling of GDP to GTP on RAS proteins. This cycling induces a conformational change that allows for binding with downstream signaling components. RAS once activated can bind RAF and signal through the mitogen-activated pathway kinase (MAPK) pathway or bind to phosphatidylinositol-3 kinase (PI3K) and signal through the PI3K pathway. The PI3K pathway can then activate or be activated by the mammalian target of rapamycin (mTOR) pathway. The downstream effect of these pathways leads to transcriptional reprogramming, affecting cellular survival, differentiation, apoptosis, and proliferation. The signaling of RAS is controlled by intrinsic GTPase activity that is catalyzed by a GTPase-activating protein (GAP) to hydrolyze the GTP into GDP and induce RAS into its inactive conformation.

**Figure 2 cancers-15-02375-f002:**
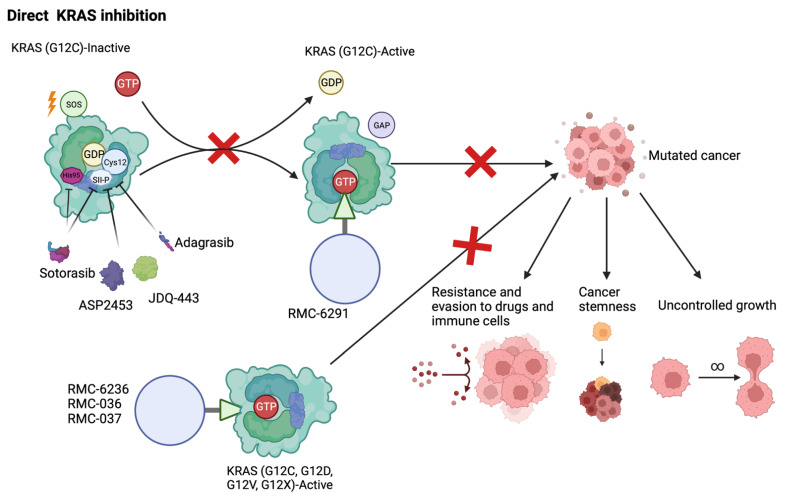
Direct inhibition of KRAS G12C mutation. The listed drugs Sotorasib, ASP2453, JDQ443, and Adagrasib work through a similar mechanism, binding to the mutated Cys12 residue of KRAS mutants. By making contacts with the Switch II pocket (SII-P) region, the inhibitors make a stable connection with KRAS and prevent structural change into its active conformation. Sotorasib uniquely binds to a His95 residue proximal to the SII-P to increase contact with the protein. In doing so, the inhibitors subdue tumorigenic phenotypes including cancer stemness, uncontrolled proliferation, and the ability to evade immune and drug attack. The series of inhibitors from Revolution Medicines, RMC-6291, RMC-6236, RMC-036, and RMC-037, bind instead to the GTP-bound active form of KRAS. Creating a tri-compound structure with Cyclophilin A, the inhibitors can bind G12C and other mutant KRAS proteins.

**Figure 3 cancers-15-02375-f003:**
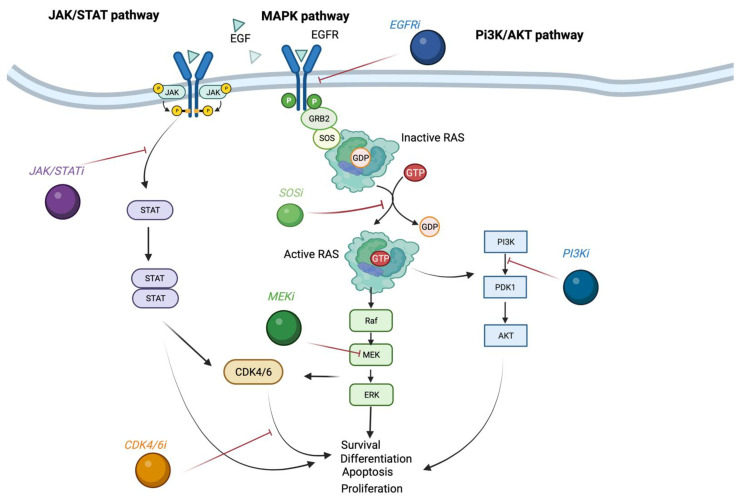
Pan-pathway inhibition. The janus kinase/signal transducer and activator of transcription (JAK/STAT) and mitogen-activated protein kinase (MAPK) pathways are both activated by a tyrosine receptor kinase such as the epidermal growth factor receptor (EGFR). The EGFR once activated leads to autophosphorylation, which in turn activates JAK proteins. The STAT protein is then activated and can activate the cyclin-dependent kinase 4/6 (CDK4/6) proteins to affect cellular physiology. In the MAPK pathway, the EGFR activates the growth factor receptor-bound protein 2 (GRB2), which acts as an adapter protein to activate the son of sevenless (SOS) protein. SOS can directly facilitate the cycling of guanine diphosphate (GDP) on the rat sarcoma virus (RAS) protein for guanine triphosphate (GTP), inducing its active conformation. Active KRAS can continue signaling through the MAPK pathway by binding the rapidly accelerated fibrosarcoma (RAF) protein, or it can act bind phosphatidylinositol-3 kinase (PI3K) and promote signaling through this pathway. RAF can then bind the MAPK/ERK kinase (MEK), which can activate extracellular signal regulated kinase (ERK). In the PI3K pathway, phosphatidylinositol-dependent kinase 1 (PDK1) is activated, which activates protein kinase B (AKT). All three pathways are distinctly different but are interconnected; the inhibition at any of one these steps can easily be compensated and therefore requires multi-targeted inhibition for effect.

**Table 1 cancers-15-02375-t001:** Current direct KRAS-mutant inhibitors.

Drug	Alternate Names	Manufacturer	Target	Development Stage
ARS-853		Wellspring Biosciences	G12C	In vivo
ARS-1620		Wellspring Biosciences	G12C	In vivo
AMG510	Sotorasib	Amgen	G12C	FDA-approved
MRTX849	Adagrasib	Mirati therapeutics	G12C	FDA-approved
MRTX-EX185		Mirati therapeutics	G12D	In vivo
MRTX-1133		Mirati therapeutics	G12D	Phase ½ Clinical trials
ASP2453		Astellas Pharma Inc	G12C	In vivo
RMC-6291		Revolution Medicines	G12C	Phase 1 Clinical trial
RMC-6236		Revolution Medicines	G12C, G12D, G12V, G12X	In vivo (not published)
RMC-036		Revolution Medicines	G12D	In vivo (not published)
RMC-037		Revolution Medicines	G12D	In vivo (not published)
BBO-8520		Bridgebio Pharma	G12C	In vivo (not published)
ERAS-3490		Erasca	G12C	In vivo (not published)
JDQ443		Novartis	G12C	Phase 1–3 Clinical trials

**Table 2 cancers-15-02375-t002:** Current clinical trials evaluating therapies of KRAS-mutant cancers with a focus on colorectal cancers.

Drug	Clinical Trial Identifier	Phase	Treatment	Cancer Type	Status and Estimated Primary Completion	Enrollment Number
Direct Inhibitors						
Adagrasib	NCT04793958	III	In combination with Cetuximab against FOLFOX or FOLFIRI	KRAS-G12C-mutant advanced/metastatic colorectal cancer	Recruiting-April 2024	420
	NCT03785249	I/II	Single agent or with Cetuximab/Pembrolizumab/Afatinib	Advanced KRAS G12C mutated solid cancer	Recruiting-December 2023	822
	NCT05375994	I/II	In combination with Avutometinib	G12C mutated NSCLC	Recruiting-July 2024	85
	NCT05578092	I/II	In combination with MRTX0902 (SOSi)	KRAS G12C mutated advanced solid cancer	Recruiting-July 2026	225
	NCT05178888	I	In combination with Palbociclib	KRAS G12C mutated advanced solid cancer	Active-December 2023	11
	NCT05472623	III	Single agent or with Nivolumab	KRAS G12C NSCLC	Not yet recruiting-November 2025	42
	NCT04613596	II	Single agent or in combination with Pembrolizumab	KRAS-G12C-mutant advanced/metastatic NSCLC	Recruiting-March 2028	950
		III	Combination with Pembrolizumab versus Pembrolizumab plus chemotherapy			
	NCT04330664	I/II	In combination with TNO155	KRAS G12C mutated advanced solid cancer	Active-September 2022	86
	NCT05722327	I	In combination with Cetuximab and Irinotecan	KRAS-G12C-mutant colorectal cancer	Not yet recruiting-September 2025	24
	NCT05609578	II	In combination with Pembrolizumab	KRAS-G12Cmutant advanced/metastatic NSCLC	Recruiting-June 2025	90
	NCT04685135	III	Single agent versus Docetaxel	KRAS-G12C-mutant advanced/metastatic NSCLC	Recruiting-August 2023	340
	NCT05634525	I	Single agent	KRAS-G12C-mutated pancreatic cancer	Not yet recruiting-November 2025	14
Sotorasib	NCT05451056	II	Single agent	KRAS-G12C-mutant NSCLC	Not yet recruiting-January 2026	37
	NCT05118854	II	In combination with Cisplatin or Carboplatin and Pemetrexed	KRAS-G12C-mutant NSCLC	Recruiting-October 2023	27
	NCT05374538	I	In combination with VIC-1911 (Aurora Kinase A inhibitor)	KRAS-G12C-mutant advanced/metastatic NSCLC	Recruiting-March 2026	140
	NCT04933695	II	Single agent	KRAS-G12C-mutant advanced/metastatic NSCLC	Active-May 2024	42
	NCT05054725	II	In combination with RMC-4630	KRAS-G12C-mutant NSCLC	Active-September 2023	47
	NCT05638295	II	Single agent or with Panitumumab	KRAS-G12C-mutant advanced/metastatic solid cancers	Not yet recruiting-December 2025	105
	NCT05198934	III	In combination with Panitumumab versus chemotherapy (Tas-102 or Regorafenib)	KRAS-G12C-mutated metastatic colorectal cancer	Active-May 2023	160
	NCT04185883	I/II	Single agent or in combination with several anti-cancer therapies	KRAS-G12C-mutant advanced solid cancers	Recruiting-July 2026	1143
	NCT03600883	I/II	Single agent or with Anti-PD1/L1 or Midazolam	KRAS-G12C-mutant advanced solid cancers	Active-May 2026	713
	NCT05480865	I	In combination with BBP-398 (SHP2 Inhibitor)	KRAS-G12C-mutant metastatic solid cancers	Recruiting-June 2024	85
	NCT05074810	I/II	In combination with Avutometinib	KRAS G12C. Mutant NSCLC	Recruiting-December 2023	53
	NCT05313009	I/II	In combination with Tarlozotinib (HER kinase inhibitor)	KRAS-G12C-mutant NSCLC	Recruiting-December 2023	30
	NCT04303780	III	Single agent versus Docetaxel	KRAS-G12C-mutant advanced/metastatic NSCLC	Active-August 2022	345
MRTX-1133	NCT05737706	I/II	Single agent	KRAS-G12C-mutant advanced solid cancers	Recruiting-August 2026	304
RMC-6291	NCT05462717	I	Single agent	KRAS-G12C-mutant advanced solid cancers	Recruiting-November 2024	117
RMC_6236	NCT05379985	I	Single agent	KRAS-G12C-mutant advanced solid cancers	Recruiting-June 2024	141
JDQ443	NCT05132075	III	Single agent versus Docetaxel	KRAS-G12C-mutant advanced NSCLC	Recruiting-April 2025	360
	NCT05445843	II	Single agent	KRAS-G12C-mutant advanced/metastatic NSCLC	Recruiting-November 2026	120
	NCT04699188	I/II	Single agent and in combination with TNO155 and/or Tislelizumab	KRAS-G12C-mutant advanced solid cancers	Recruiting-May 2025	375
	NCT05714891	II	Single agent	Surgically resectable NSCLC	Not yet recruiting-August 2026	27
	NCT05358249	I/II	In combination with Trametinib, Ribociclib, or Cetuximab	KRAS-G12C-mutant advanced solid cancers	Recruiting-June 2025	346
IBI351	NCT05497336	I	Single agent or in combination with Cetuximab	KRAS-G12C-mutant advanced/metastatic colorectal cancer	Recruiting-August 2023	80
JAB-21822	NCT05194995	I/II	In combination with Cetuximab	KRAS-G12C-mutant colorectal, small intestine and appendiceal cancer	Recruiting-December 2023	62
	NCT05288205	I/II	In combination with JAB-3122 (SHP2i)	KRAS-G12C-mutant advanced solid cancers	Recruiting-March 2026	124
GDC-6036	NCT04449874	I	Single agent or in combination with several anti-cancer therapies	KRAS-G12C-mutant advanced/metastatic solid cancers	Recruiting-November 2024	498
HBI-2438	NCT05485974	I	Single agent	KRAS-G12C-mutant advanced solid cancers	Recruiting-August 2025	44
ELI-002	NCT05726864	I/II	Single agent	KRAS/NRAS-mutated solid cancers	Recruiting-November 2026	156
	NCT04853017	I	Single agent	KRAS/NRAS-mutated solid cancers	Recruiting-January 2023	18
BPI-421286	NCT05315180	I	Single agent	KRAS-G12C-mutant and not advanced solid cancers	Recruiting-July 2023	80
DCC-3116	NCT04892017	I/II	Single agent or in combination with Trametinib, Binimetinib, or Sotorasib	RAS/MAPK-mutant advanced/metastatic solid cancers	Recruiting-April 2024	323
BDTX-4933	NCT05786924	I	Single agent	BRAF-mutant, select KRAS and NRAS-mutant advanced/metastatic solid cancers	Recruiting-June 2026	140
Indirect targeting						
RMC-4630	NCT04916236	I	In combination with LY3214996	KRAS-mutant metastatic solid cancers	Recruiting-January 2024	55
Vitamin C	NCT03146962	II	Single agent	RAS/BRAF-mutant and wildtype advanced/metastatic solid cancers	Recruiting-June 2023	78
Telaglenastat	NCT03965845	I	Single agent or in combination with Palbociclib	KRAS-mutant advanced solid cancers	Completed-September 2021	53
Immunotherapy						
Sintilimab	NCT04745130	II	In combination with Regorafenib and Cetuximab	Metastatic colorectal cancer (KRAS wt and mutant)	Recruiting-February 2023	90
KRAS peptide vaccine	NCT04117087	I	In combination with Nivolumab and Ipilumab	KRAS-mutant PDAC or metastatic colorectal cancer	Recruiting-December 2024	30
	NCT03953235	I/II	In combination with Nivolumab and Ipilumab	RAS/BRAF/TP53-mutant MSS-CRC, NCLC, PDAC	Active-December 2023	30

**Table 3 cancers-15-02375-t003:** Current molecules and methods evaluated to increase the response of immunotherapy in colorectal cancer.

Molecule/Therapy	Mechanism to Aid Immunotherapy
Blocking interleukin 17A (IL-17A)	Decreases PD-L1 expression and increases cytotoxic T-cells
Degrading Tribbles homolog 3 (TRIB3)	Increases tumor lymphocyte infiltration
Degrading neutrophil extracellular traps (NETS)	Removes a physical barrier between immunotherapy and cancer cells
Introducing mitochondrial antiviral signaling gene (MAVS)	Increases CD8 T-cells cytotoxicity, and the expression of PD-L1
5-FU + oxaliplatin combination	Induces immunogenic cell death that stimulates the immune system
Tas-102 + oxaliplatin combination	Induces immunogenic cell death that stimulates the immune system
PF-07062119 treatment	Bispecific antibody that acts as a scaffold to bring T-cells in close proximity to cancer cells also increases PD-1 and PD-L1 expression
Adoptive T-cell transfer	Expands cancer cell specific T-cells ex vivo for treatment
Sotorasib	Increases T-cell priming and infiltration, antigen presentation, and activation
Vaccine	Primes immune system to KRAS-mutant proteins for adaptive killing

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
