# Peer review of "The Therapeutic Landscape for KRAS-Mutated Colorectal Cancers"

_cancers, 2023, doi:10.3390/cancers15082375_

Round 1

Reviewer 1 Report

Activating mutations of the small GTPase K-RAS are very found in many cancers and are the driving mutations of ~40% of metastatic colorectal cancers. Inhibitors developed for downstream oncoproteins such as Raf and MEK are usually not efficient in curing mutants K-Ras containing tumors. Indeed, patients baring these mutations have been traditionally difficult to manage. In order to overcome this problem, a great effort has been invested over the past years to develop inhibitors targeting KRAS mutations. Indeed, quite a few inhibitors are now in development, and two (Sotorasib and Adagrasib), directed specifically towards the G12C mutant, are already in clinical use. Although these drugs are initially effective, all patients develop drug resistant within a few months. This can be solved, at least in part, by combining the Ras inhibitors with immuno-therapy, drugs directed to the resistance-inducing pathways, or other approaches.  

In this review the authors nicely describe the recent development in this important field, particularly on the clinical side. It is a well-written and timely review that covers many important aspects of K-Ras inhibition in colorectal cancer. I have only minor corrections and suggestions for the authors’ consideration. 

Comments:

1)  Fig. 1: The presentation of Ras activation is somewhat misleading. It should be made clearer that the RasGDP (that appears in the right side) is the target of SOS that releases the GDP. Here it looks like Ras is activated by its GTPase activity. In addition, it would be beneficial to include mTORC2 and maybe also PTEN in the PI3K pathway.  

2)  Fig. 2: The sequence of events, (although correct in principle) is difficult to understand. Can the author split the scheme into two parts. One that describes the normal activation of Ras and the second that shows the effect of the drugs. 

3)  Fig. 3: Again, the mode of Ras activation is difficult to understand (as in Fig. 1). In addition, it is not clear why the inhibitors of Raf, ERK, PDK, mTOR, and AKT are not mentioned. 

4)  It is suggested to describe in more details other oncogenic mutations of KRas (not only G12C) and their mechanism of action. 

5)  The authors mention that 25-52% of all colorectal cancer patients have KRas mutations (line 85). What is the reason for this wide range?

6)  I suggest to describe the molecular mechanism of action of Sotorasib and Adagrasib in more details (covalent binding is not mentioned for Sotorasib). What is the reason that Adagrasib is more effective than Sotorasib?

7)  Raf and MEK inhibitors acquire resistance by a large number of mechanisms (more than 50 distinct ways have been described). Here it seems that the number of mechanisms is smaller. What is the reason? Can the authors speculate why no activating mutations of Raf have been identified as a way the develop resistance. 

8)  The section describing immunotherapy is not so clear. How do the Ras inhibitors sensitize the cells to immunotherapy? 

Author Response

Answers will be in the word document

Reviewer 2 Report

A timely article by Drs Whitehall and Tria discusses the therapeutic opportunities of KRAS-mutated colorectal cancer, which is much needed since this field is moving as fast as ever moving field after KRAS G12C inhibitors have been FDA-approved. This review is a very well-written document and has been updated with recent articles. Though a few things need to be addressed before it is ready for acceptance, they are as follows:

1. It has been discussed that KRAS-driven colorectal cancer offers therapeutic opportunities by exploiting the altered metabolism (PMID: 26541605 and PMID: 33870211). Authors must add a few lines discussing this topic.

2. Table 1 must b updated; a few more inhibitors information are available to the public. Please include those. For example, check BBO-8520 and ERAS-3490, including several others. 

3. Authors should add a table describing the current clinical trials in KRAS mutant colorectal cancers with related details information. 

Author Response

answers will be in word document

Round 2

Reviewer 2 Report

The authors almost addressed all concerns, and now the paper looks improved and better.  Only one minor edit must be done before this paper gets accepted. Authors should refer to two recent reviews in KRAS and the metabolism field while discussing the topic in the text. This has been mentioned in the previous comments, but probably authors somehow overlooked it. This way, the references will be updated with the manuscript. Please refer to (PMID: 34244683 and PMID: 33870211) between Ref 99 to 104, where KRAS and metabolism have been discussed in the text.  

Author Response

- The requested references have been added on page 13, lines 460 and 487